# Widespread loss of safe lake ice access in response to a warming climate

**Joshua Culpepper**[1]*, **Lei Huang**[2,3], **R. Iestyn Woolway**[4], **Sapna Sharma**[5]

**1** Department of Biology, York University, Toronto, ON, Canada, **2** College of Resource Environment and Tourism, Capital Normal University, Beijing, China, **3** Institute for Basic Science, Center for Climate Physics, Busan, South Korea, **4** School of Ocean Sciences, Bangor University, Bangor, United Kingdom, **5** Department of Biology, York University, Toronto, ON, Canada

* joshua.abel.culpepper@gmail.com

**Data Availability Statement:** Data from the CESM2 Large Ensemble Community Project are publicly available at: https://www.cesm.ucar.edu/projects/community-projects/LENS2/ data-sets.

## Abstract

Millions of people rely on lake ice for safe winter recreation. Warming air temperatures impact the phenology (timing of formation and breakup) and quality (ratio of black to white ice) of lake ice cover, both critical components of ice safety. Later formation and earlier breakup of lake ice lead to overall shorter periods of use. However, greater proportions of white ice may further inhibit safe ice use owing to its lower weight-bearing capacity. As ice cover duration decreases and ice quality changes in a warming world, the period of safe ice use will similarly diminish. We use a large ensemble modeling approach to predict ice safety throughout the winter period in the Northern Hemisphere. We used the Community Earth System Model Version 2 Large Ensemble (CESM2-LE) to calculate the period when ice first appears until it is of sufficient thickness for safe use, which depends on the ratio of black to white ice. We conducted this analysis for 2,379 to 2,829 ~1˚ by 1˚ grid cells throughout the Northern Hemisphere. We focus on the period between ice formation ($\geq$ 2 cm) to a safe thickness for general human use (i.e., $\geq$10, $\geq$15, or $\geq$20 cm, depending on the ratio of black to white ice). We find that the transition period from unsafe to safe ice cover is growing longer, while the total duration of safe ice cover is getting shorter. The transition period of unsafe ice increases by 5.0 ± 3.7 days in a 4˚C warmer world, assuming 100% black ice. Diminished ice quality further limits safe ice conditions. The unsafe transition period increases by an average of 19.8 ± 8.9 days and 8.8 ± 6.6 days for the ice formation and breakup periods, respectively in a 4˚C warmer world assuming 100% white ice conditions. We show that although many lakes are forecasted to freeze, they will be unsafe to use for an average of 5 to 29 fewer days in a 4˚C warmer world for 100% black and 100% white ice ratios, respectively. This wide range indicates that ice quality has a strong influence on ice safety. This work highlights the need to understand both lake ice phenology and quality to better assess safe lake ice use during the formation and melt periods.

## 1. Introduction

Anthropogenic climate change is causing rapid loss of lake ice in the Northern Hemisphere, driven largely by rising air temperatures [1]. Studies have illustrated that the timing of ice-on

html. Derived CESM2-LE data for this analysis are available at https://doi.org/10.6084/m9.figshare.26882467.v1 Code to reproduce the figures is available at https://github.com/jculpepper7/ice_safety_2024.

**Funding:** We would also like to thank the Natural Sciences and Engineering Research Council Discovery Grant, the York University Research Chair programme, and ArcticNet, a Network for Centres of Excellence Canada to SS for providing funding to support this research. RIW was supported by a UKRI Natural Environment Research Council (NERC) Independent Research Fellowship [grant number NE/T011246/1]. Lei Huang was supported by the National Natural Science Foundation of China (No. 42201049) and the Second Tibetan Plateau Scientific Expedition and Research (STEP) (grant number 2019QZKK0202). The CESM2-LE simulations presented here have been conducted through a partnership between the Institute for Basic Sciences (IBS) Center for Climate Physics (ICCP) in South Korea and the Community Earth System Model (CESM) group at the National Center for Atmospheric Research (NCAR) in the United States, representing a broad collaborative effort between scientists from both centers. The simulations were conducted on the IBS/ICCP supercomputer "Aleph." The funders had no role in study design, data collection and analysis, decision to publish, or preparation of the manuscript.

**Competing interests:** The authors have declared that no competing interests exist.

and ice-off (ice phenology) is changing around the Northern Hemisphere. Lake ice duration has been a central concern, where long-term records have shown an overall increase in open-water days of 0.62 days per decade between 1931–2005, with a nonlinear increase in the rate of open-water days in the last 30 years of the time series [2]. Lake ice phenology through long-term records extending over a century (1846–2019) show ice formation to be 11 days later and ice breakup to be 6.8 days earlier, which was a near doubling of ice loss compared to the same lake ice records between 1846–1995 [3]. Moreover, lake ice is thinning at a rate of 0.033 m per C increase in air temperature [1]. Therefore, long-term lake ice records show an overall loss of ice cover duration, timing, and thickness directly related to warming air temperatures [1,4].

However, climate change and warming winters have farther reaching effects by also degrading lake ice quality conditions [5,6]. Ice quality depends in large part on winter air temperatures, where warm temperatures induce diel melt-freeze cycles and/or precipitation to fall as both snow or rain. These temperature and precipitation conditions lead to white ice formation from freezing slush that introduces many air bubbles and small ice crystals that reflect light [7,8]. White ice is also notable for being structurally weaker than black ice [9]. The closer to 0˚C the air temperature is, the weaker the ice becomes [10]. For example, white ice at -0.5˚C is 51% weaker than black ice at the same temperature, owing to the increased strength of ice with increased density [10]. White ice formation is generally expected toward the end of the season when temperatures warm and snow atop the ice column leads to a larger fraction of white ice as the ice column evolves throughout the season toward breakup [5]. However, white ice conditions are possible at the beginning of the season as well when ice formation and snow conditions coincide to form larger proportions of white ice [11]. Therefore, warming conditions expected throughout the 21st Century may lead to a more consistent presence of white ice throughout the season.

Deteriorating ice conditions pose a significant safety hazard for human activities on frozen lakes. Freshwater ice plays a critical role in human winter mobility across the Northern Hemisphere [12] spanning millions of lakes [13]. Warmer air temperatures, however, are linked to increased search-and-rescue events as well as non-fatal and fatal ice-related drownings [14,15]. While the majority of drownings occur during seasonal transitions when ice is weakest (beginning and end of the ice season) [15], even the peak ice season can pose dangers if ice quality is compromised. This was evident in the February 2021 tragedy in Sweden, where fatalities occurred despite seemingly sufficient ice thickness due to the absence of the stronger "black ice" layer and the sole presence of weaker "white ice" [5].

In this study, we build on earlier global lake ice research [1,16–18] by investigating the influence of climate warming and ice quality on the duration of the transition period between unsafe and safe ice conditions on Northern Hemisphere lakes. Safe ice conditions are defined by a combination of thickness and ice quality, where 10 cm of black ice or 20 cm of white ice are required for a person to safely walk on the ice [5]. By incorporating the period of unsafe ice cover in the analysis, this work adds vital context to ice phenology studies and directly addresses the impact of changing climate on ecosystem services, specifically pertaining to safe ice use. We aim to address three key questions: 1) how will the timing of safe ice availability change during various climate warming scenarios (1˚C, 2˚C, and 4˚C), 2) how will different ice quality conditions (100% black ice to 100% white ice) interact with warming climates, and 3) what are the potential consequences of changing ice phenology on safe ice use? We anticipate a lengthening unsafe transition period for both ice formation and melting due to warming air temperatures impacting the overall ice cover season, ice thickness, and ice quality. We posit that a slower ice development at the beginning of winter will delay the period of safe ice availability, while rapid spring warming will hasten the breakdown of the ice cover below safe thickness levels. Furthermore, diminished ice quality is expected to extend the unsafe period due to the requirement of greater ice thickness for safe weight support.

## 2. Methods

### 2.1 Modeled data

We employed the lake simulation results in the Community Earth System Model Version 2 Large Ensemble (CESM2-LE) project to estimate phenological shifts in lake ice thickness across the Northern Hemisphere [19]. Lake ice thickness was derived from the Lake Ice, Snow, and Sediment Simulator (LISSS), which is a one-dimensional process-based model that incorporates ice growth, with the additional complexities of aerosol deposition on snow albedo, freeze-thaw cycles, ice physics, and sediment heat flux throughout the water column [20]. The model is integrated at a 30-min time step and forced by meteorological variables, including air temperature, pressure, snow, humidity, wind speed, shortwave radiation, and downward longwave radiation [21] from the atmospheric model in the CESM2. We used daily mean of modeled ice thickness to detect the day of the year when ice growth reached a specified thickness deemed safe. We then compared the temperature anomalies at 1˚C, 2˚C, and 4˚C to a historical climate period of 1851–1880, representing a 30-year period of unperturbed climate by human activities. For a more thorough explanation of the lake ice variables derived from CESM2-LE and LISSS, see [16,20].

### 2.2 Validation

The spatial extent and resolution of the study include the Northern Hemisphere on a 0.9˚-by-1.25˚ gridded scale. Within each grid, the LISSS coupled to the CESM2-LE uses a sample lake to model ice thickness, where the sample lake has a depth and surface area of the mean of individual lakes within the grid, taken from the Global Lake and Wetland Database [22] and global gridded depth data at a 1 km resolution [23]. CESM2-LE offers data for lake ice-on and ice-off timing, using the daily lake ice thickness from the one-dimensional lake ice model. The LISSS model has shown accurate representation of ice thickness and other variables (e.g., water temperature and surface fluxes) in previous studies [16,20,24–27]. Ice thickness was validated with records from 15 Canadian lakes with an $R^2 = 0.86$. While the correlation coefficient was high, the modeled data deviated from the 1:1 line, such that the model tended to underpredict ice thickness (slope = 2.1) [16]. This bias toward thinner ice cover values is likely due to the warm bias of the simulated land surface air temperature in the fully coupled model, which was warmer than the ERA5 reanalysis data between the study years 1981–2020 and 30˚N-70˚N latitude by 1.5˚C [16]. Additional validation information can be found in [16] and the associated supplementary information.

Owing to the small sample size in [16], in this study we sought to further validate the ice thickness component of LISSS. We aggregated 71 lakes from datasets from Finland [28], North America [29,30], and Russia [31]. Using 71 ice thickness records with observations between 1980-present, we found that a comparison of observed and modeled ice thickness yielded a root mean square error (RMSE) of 18.5 cm and an $R^2 = 0.74$ (n = 71) (Fig 1). As with the validation procedure in [16], the modeled data tended to underpredict ice thickness, which is evident from the slope of 0.49 (Fig 1A). When directly examining lake ice thickness time series, the mean ice thickness of the ensemble members tends to underestimate the peak ice thickness (Figs 1B and S1). The mean and $2\sigma$ may also underpredict the peak ice thickness; however, they tend to capture the ice growth period and ice loss period, which are most important for this study. The model also tends to predict lake ice every year (1980–2024); however, some lakes did not freeze in years during data collection. It should be noted that the lake ice thickness model (LISSS) uses a mean lake area and depth to model ice thickness [16,20]. Therefore, peak lake ice thickness is likely to vary between lakes that do not approximate that mean area and depth.

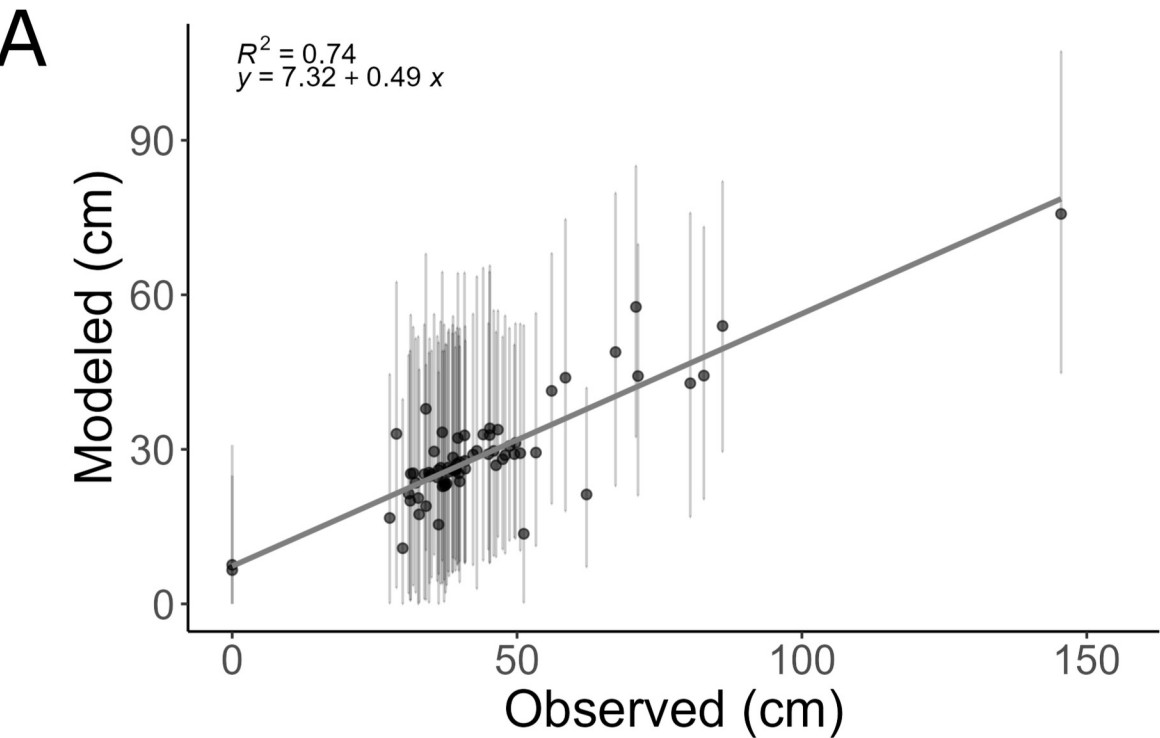

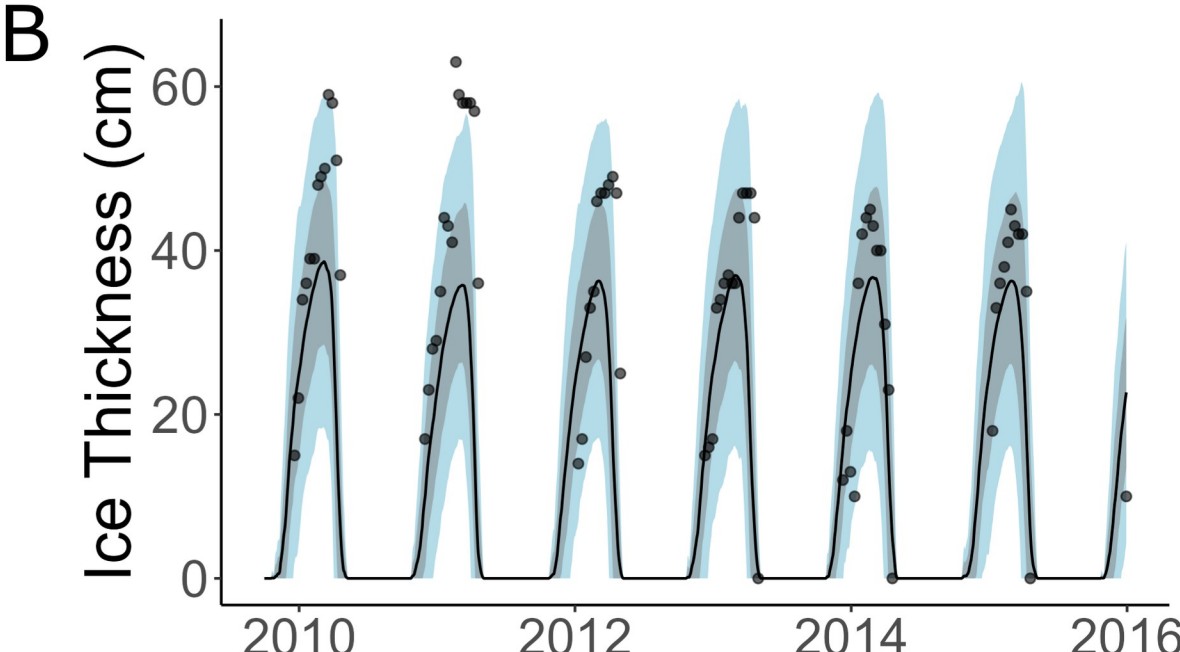

**Fig 1. Observational and model data comparison.** Figure shows a) the comparison between modeled and observed data, where points are the 71 individual lakes and error bars are 2σ of the 80 ensemble members used to derive the mean daily values. b) A portion of an example time series with the black line representing the mean of the 80 ensemble members for a grid that incorporates the lake and points, which are observations. The grey and blue ribbons represent 1σ and 2σ, respectively.

## 2.3 Determining ice quality

Ice quality is an often overlooked and understudied metric due to a paucity of data but have significant impacts on ice safety [5]. For example, the flexural strength of white ice is approximately 51% lower than black ice at temperatures near 0˚C [10] and the load bearing capacity of lake ice can drop significantly when ice quality is low [9]. It is important to note that the CESM2-LE does not provide a ratio of black ice and white ice, though it does incorporate freeze-thaw cycles, snow grain evolution, and snow compression into the ice column [20,32]. Therefore, in order to make use of ice quality as a component of ice safety, we combined safety protocols from the Minnesota Department of Natural Resources (MDNR) to determine allowable loads on black ice and white ice [5,33]. The ice quality gradients in this study focus on assumed scenarios of lake ice at 100% black ice, 100% white ice, and a 50% black ice and 50% white ice scenario.

We defined the transition period from unsafe ice to safe ice thickness as the duration of time from ice first appearing at a thickness of 2 cm (i.e., ice on) to ice reaching 10 cm, 15 cm, or 20 cm, depending on the hypothetical ice quality. We chose 2 cm as ice on to eliminate short-lived, intermittent ice events that might occur if defining ice as >0 cm. For example, the initial skim ice that forms on a calm, cold night can break up easily from wind stress or melt from warmer temperatures during the day [7]. These periods of brief ice cover are not typically considered as the date of ice formation in ice phenology records and are known to add uncertainty to lake ice records [34]. Using an ice thickness > 2 cm permitted us to consider the ice column of each lake within a grid to be stable. For pure black ice, a thickness of 10 cm was used to conform with the MDNR safety recommendations for a person walking on ice. For 50% white ice and 100% white ice, we used 15 cm and 20 cm of ice thickness, as the thickness is doubled for white ice compared to black ice for safe use [5]. Therefore, we used the equivalent thickness required for safe ice use, owing to the inability of the model to distinguish between black and white ice layers directly. We extracted the day of the year when the ice reached the specified thicknesses for a historical climate (1851–1880), 1˚C, 2˚C, and 4˚C warming. These temperature thresholds were chosen to cohere with an established framework of indexing global temperature rise instead of a particular emission scenario since lake ice cover is strongly influenced by the global mean surface temperature [35]. We also excluded grids where ice cover lasted for fewer than 30 days or longer than 330 days. We then determined the phenological anomaly by subtracting the warming scenarios from the historical climatology for black ice for each individual 0.9˚-by-1.25˚ grid. From those gridded values, we then took the Northern Hemisphere average and standard deviations.

## 2.4 Statistical analysis

We tested each grouping of temperature and ice quality for normality using a Shapiro-Wilks test (S1 Table). We then tested for significant differences in transition periods across warming scenarios within an ice quality category and across ice quality categories within the same warming scenario by using a Kruskal-Wallis test (S2 and S3 Tables). If the means differed significantly ($p < 0.05$) between the three groups, we used a Dunn's test to determine which means were significantly different (S4 and S5 Tables). We verified the family-wise error rate using the Holm method [36]. Subsequently, we used a Kolmogorov-Smirnov two-sample test to determine whether distributions were significantly different, again both between warming scenarios within the same ice quality category and across ice quality categories within the same warming scenario (S6 and S7 Tables). We used the KS test to determine the absolute difference between the empirical cumulative distribution functions of the different distributions. The

Shapiro-Wilk test, Kruskal-Wallis test, and the Kolmogorov-Smirnov test were conducted in the 'stats' package in R [37]. Dunn's test used the 'FSA' package in R [36,38].

## 3. Results

### 3.1 Transition period timing

Ice formation at a thickness of 2 cm occurred on day 313 (i.e., November 9th) on average during the historical reference period (1851–1880) across the Northern Hemisphere. Ice breakup occurred on approximately day 121 (i.e., May 1st). After ice formation, ice growth continued and reached a safe thickness for general ice safety (i.e., walking on ice) approximately 6 days later, during the historical reference period. This duration between the average formation of ice across the Northern Hemisphere and the increase of ice thickness to a stable ice cover for use is the transition period between unsafe and safe ice (Fig 2). This 6-day transition period stipulated that the ice column is 100% black ice, which supports more weight with a smaller thickness than does white ice [5]. During the same period, it took 19 days from ice formation ($\geq$ 2 cm) to a 20 cm ice thickness, which is a required thickness for general ice use under 100% white ice conditions. Ice breakup occurred over a more compressed period, where lake ice diminished from 20 cm (i.e., 100% white ice conditions) to a complete ice breakup (< 2 cm) over only 4 days.

### 3.2 Quantifying increasing transition periods

Both anticipated warming air temperatures and ice quality degradation interacted to prolong the duration of the transition period from unsafe to safe ice during the ice formation period (Fig 2). The transition period during ice formation increased in duration by 1.1±2.3 (SD, n = 2829), 1.5±2.7 (SD, n = 2762), and 2.6±3.7 (SD, n = 2618) days when air temperatures warmed 1˚C, 2˚C, and 4˚C, respectively (Table 1). These increases imply that lake ice takes longer to increase in thickness from initial ice formation ($\geq$ 2 cm), which is unsafe to walk on, to a safe thickness ($\geq$10 cm, $\geq$15 cm, and $\geq$20 cm, depending on assumed ice quality). Simultaneously, lake ice formed later. The duration of the transition period increased further when considering ice quality (Fig 3). The transition period increased by 8.0±4.9 (SD, n = 2743), 8.7±5.7 (SD, n = 2674), and 10.7±6.3 (SD, n = 2512) days when the ice column changes to 50% white ice and air temperatures warm by an additional 1˚C, 2˚C, and 4˚C, respectively, compared to historical black ice conditions (Table 1). Finally, when the lake ice column was entirely composed of white ice, the transition period duration increased by 15.3±8.1 (SD, n = 2643), 16.6±8.6 (SD, n = 2580), and 19.8±8.8 (SD, n = 2379) days when air temperatures warmed 1˚C, 2˚C, and 4˚C, respectively, compared to historical black ice conditions (Table 1). The increasing duration of the transition period between unsafe and safe ice during the ice formation process implies that lake ice, on a Northern Hemisphere average, can be delayed from November 15th to between December 13th and January 18th. This date range incorporates both the average later ice formation and the average transition period with their relative standard deviations. This date range, then, incorporates the minimum and maximum projected change in the timing of safe ice thickness.

The transition period followed a similar, though muted, pattern during the melt period when incorporating the compound effects of increasing air temperatures and ice quality. The unsafe transition period during the lake ice melt occurs when lake ice thins to $\leq$10 cm (black ice), $\leq$15 cm (50% black ice and 50% white ice), and $\leq$20 cm (white ice) until ice breakup ($\leq$2 cm) (Fig 2). This transition period extended under warming of 1˚C, 2˚C, and 4˚C by 0.81±2.4 (SD, n = 2829), 1.4±2.6 (SD, n = 2762), and 2.4±3.6 (SD, n = 2618) days. When combined with 50% white ice, the transition period increased to 3.9±3.5 (SD,

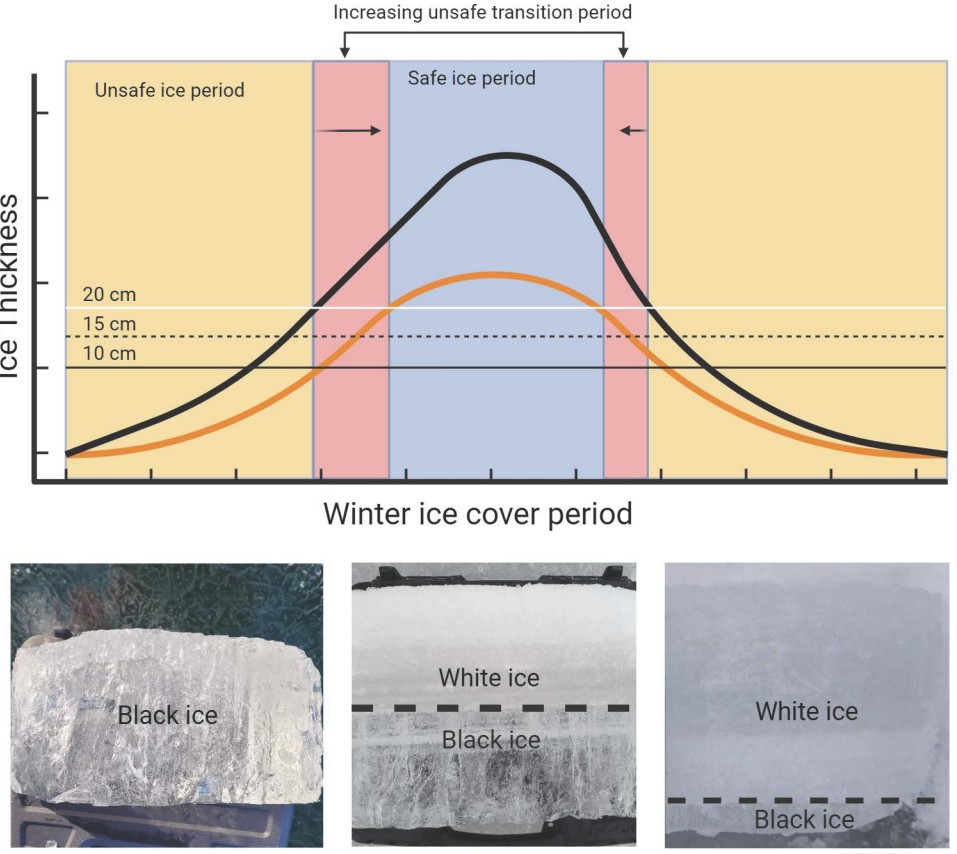

**Fig 2. Unsafe transition period conceptual model.** The conceptual model depicts the thickening of lake ice after formation for a historical period (black line) and a future warming scenario (orange line). The horizontal lines indicate the required thickness that the ice must reach prior to use under different ice quality categories (black ice: >10 cm (solid black line), a mix of black and white ice: 15 cm (dashed line), and white ice: >20 cm (white line)). The blue shaded area represents the period of safe ice use. The yellow shaded area represents the period of unsafe ice use for the historical period, while the red region represents the increasing length of the ice cover period for the future warming scenario. For this instance, the red shaded region is the difference between the transition period for white ice for the warming scenario and the historical black ice scenario. The panels at the bottom show various ice categories as sampled from Paint Lake and Lake Simcoe during the winter of 2023–2024. From left to right, they are 100% black ice, 50% black ice and 50% white ice, and almost entirely white ice.

n = 2743), 4.7±4.4 (SD, n = 2674), and 5.9±5.4 (SD, n = 2512) days when average air temperature warms by 1˚C, 2˚C, and 4˚C, respectively. Finally, when ice quality was 100% white ice, the transition period extended by 6.3±4.9 (SD, n = 2643), 7.2±5.5 (SD, n = 2580), and 8.8±6.6 (SD, n = 2379) days under warming of 1˚C, 2˚C, and 4˚C, respectively (Table 1, Fig 3).

Additionally, the standard deviation of the transition period increases with warming scenarios and across ice quality ratios, which implies that the variability of the transition period increases with both warming and degrading ice quality (Table 2). For example, the standard deviation of black ice during the ice formation period increases from 11.5 to 13.3 days between the historical period and 4˚C. The standard deviation of white ice, on the other hand, increases from 23.7 to 30.0 days between the same period. The standard deviation during the melt period is overall shorter compared to the ice formation period; however, the values increase with both warming and degrading ice quality (Table 2).

**Table 1. Unsafe transition period anomalies.**

| Warming | Formation Transition Anomaly | Formation Transition Standard Deviation | Melt Transition Anomaly | Melt Transition Standard Deviation | Sample size |
|---|---|---|---|---|---|
| **100% Black Ice** | | | | | |
| **1˚C** | 1.1 | 2.3 | 0.8 | 2.4 | 2829 |
| **2˚C** | 1.5 | 2.7 | 1.4 | 2.6 | 2762 |
| **4˚C** | 2.6 | 3.7 | 2.4 | 3.6 | 2618 |
| **50% White Ice** | | | | | |
| **1˚C** | 8.0 | 4.9 | 3.9 | 3.5 | 2743 |
| **2˚C** | 8.7 | 5.7 | 4.7 | 4.4 | 2674 |
| **4˚C** | 10.7 | 6.3 | 5.9 | 5.4 | 2512 |
| **100% White Ice** | | | | | |
| **1˚C** | 15.3 | 8.1 | 6.3 | 4.9 | 2643 |
| **2˚C** | 16.6 | 8.6 | 7.2 | 5.5 | 2580 |
| **4˚C** | 19.8 | 8.9 | 8.7 | 6.6 | 2379 |

The duration anomaly of the "shoulder season" when ice is $\geq 2$ cm and $\leq 10$ cm (black ice), $\leq 15$ cm (50% white ice), or $\leq 20$ cm (white ice), relative to a base period of 1851–1880. Values in parentheses are standard deviation. These values are for the Northern Hemisphere (latitude > 40 & < 80). Warming is in degrees Celsius; transition periods and standard deviation are in days and sample size is the number of grids present in the analysis.

### 3.3 Spatial variation

Transition periods show distinct regional variability throughout the Northern Hemisphere but overwhelmingly shift toward longer unsafe ice transition periods. Unsafe transition periods on average increased (Table 1; Fig 3) and transition periods tended to increase the most along coastal regions in North America as well as southern latitudes in North America, Europe, and Asia. For example, transition periods exceeding 10 days composed approximately 30% of grids (n = 661) for white ice conditions at 4˚C. On the other hand, grids shifting to longer safe ice periods consisted of less than 1% of grids (n = 14). More grids shift toward longer safe ice periods at 1˚C under black ice conditions (18%, n = 496). These grids are primarily located in northern latitudes of North America, Europe, and Siberia as well as some grids along the western coast of Greenland.

### 3.4 Distribution patterns

While the transition period of both ice formation and ice melt was similar for 100% black ice, with increasing ratios of white ice within the ice column, the mean transition period became longer for the ice formation period than for the ice breakup period. The distributions of the transition period anomalies became more exaggerated and skewed when both warming and ice quality interacted, such that both the mean transition period and the distributions of the transition periods were significantly different between warming scenarios within the same ice quality category (p < 0.05) and between ice quality categories within the same warming scenario (p < 0.05).

Generally, the distribution of transition period anomalies was more constrained under black ice conditions, though it showed right skewing when warming reached 4˚C (Fig 4). However, when ice quality became 50% and 100% white ice, the right skewness of the distributions became more and more exaggerated (i.e., the distribution includes larger anomaly values). The distributions between ice quality categories are significantly different from one another. While all distributions for both ice formation and melt periods are significantly

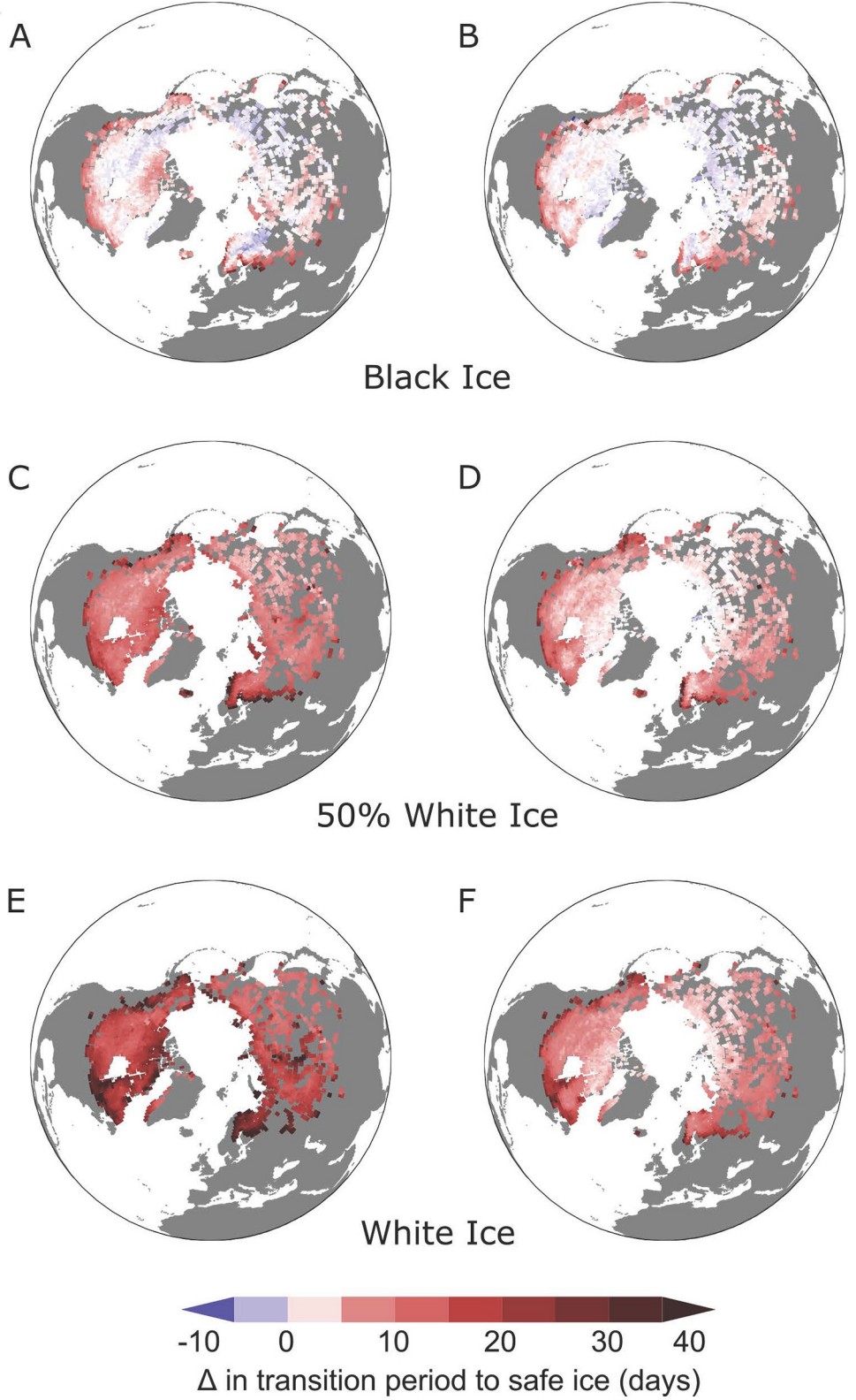

**Fig 3. Spatial variation of unsafe transition periods.** Transition periods from ice formation to safe ice thickness for the ice formation period (a,c,e) and the melt period (b,d,f) under 4°C warming, using the CESM2-LE modeled ice thickness data. The maps include transition periods for equivalent thicknesses to safely traverse 100% black ice (a,b), 50% black ice and 50% white ice (c,d), and 100% white ice (e,f).

**Table 2. Unsafe transition period standard deviation.**

| Warming | Ice Formation Standard Deviation | Ice Melt Standard Deviation |
|---|---|---|
| **100% Black Ice** | | |
| **0˚C** | 11.5 | 7.9 |
| **1˚C** | 12.2 | 8.3 |
| **2˚C** | 12.4 | 8.6 |
| **4˚C** | 13.3 | 9.1 |
| **50% White Ice** | | |
| **0˚C** | 17.3 | 10.1 |
| **1˚C** | 18.7 | 10.8 |
| **2˚C** | 19.3 | 11.4 |
| **4˚C** | 21.1 | 12.4 |
| **100% White Ice** | | |
| **0˚C** | 23.7 | 12.1 |
| **1˚C** | 25.8 | 13 |
| **2˚C** | 27 | 13.6 |
| **4˚C** | 30 | 15 |

The standard deviation of the transition periods shown for each warming scenario and ice quality ratio. Values are in units of days. Warming is in degrees Celsius. The value of 0˚C is the historical period of 1851–1880.

different ($p < 0.05$), distributions between ice quality categories had higher absolute difference values. When combined with warming, the absolute difference between 1˚C black ice and 4˚C white ice exceeded 0.99, indicating a nearly complete difference between the empirical cumulative distribution functions of the two distributions (S3 and S4 Figs). This stark variation of the distributions and empirical cumulative distribution functions (ECDF) implies that ice quality increased the transition period of unsafe lake ice more significantly than warming, with higher maximum values. That is, the period of unsafe ice presence is projected to be much longer when ice quality degrades.

# 4. Discussion

This research investigated how projected warming average air temperatures and changing ice quality throughout the 21st Century will result in longer periods when lake ice cover is present but not safe for human use. Studies in winter limnology have primarily focused on changing phenology and duration [3] and decreasing ice thickness [1,4]. Using CESM2-LE, our projections suggest a shift toward lake ice that will be unsafe for human use for a mean of approximately 5, 17, or 29 days longer per year for black ice, 50% white ice, and 100% white ice, respectively, at 4˚C of warming. We determined that ice quality increased the transition period of unsafe ice more than did warming air temperatures alone.

## 4.1 Lake ice is projected to become increasingly unsafe

Ice quality impacts the carrying capacity of lake ice, which interacts with air temperatures both in the creation of white ice [5,7] and its flexural strength [9,10]. As density decreases with degrading ice quality, lake ice can support less weight and becomes more than 50% weaker at temperatures near 0˚C [10]. Ice quality can quickly degrade in warm years, where most or all of the ice column can become white ice throughout the ice cover season, resulting in dangerous ice conditions, including drownings [5,15]. While the duration of safe lake ice use has been analyzed for transportation efforts [21], the unsafe transition period adds important

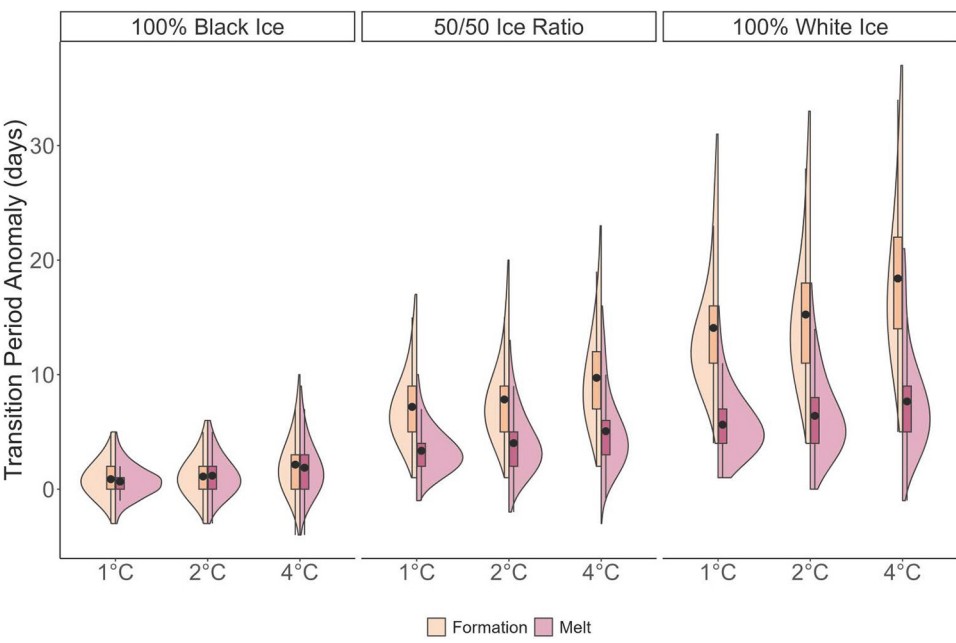

**Fig 4. Distributions of unsafe transition periods.** A split violin plot that illustrates the distribution of the transition period (in days) for warming scenarios of 1˚C, 2˚C, and 4˚C. The yellow shading demonstrates the ice formation period and the red shading represents the melt period. The boxplot within each distribution shows the quantiles, median (black point), and outlier values. Each grouping of distributions represents the ice quality for each set of warming scenarios, from left to right: Black ice, a 50/50 ice scenario, and white ice. Each average value was statistically significant (Kruskal-Wallis, p < 0.05, see S2 and S3 Tables for more details) from one another within ice quality categories and across ice quality categories for both ice formation and melt. Additionally, each distribution was significantly different from one another (Kolmogorov-Smirnov, p < 0.05, see S6 and S7 Tables for more details) across all ice quality categories and warming scenarios.

context because people often use lake ice in the early winter and late spring for recreation, when a majority of incidents take place [15].

As with ice phenology and duration, air temperature plays a large role in ice cover development and melt. Air temperature is the primary driver of lake ice formation, growth, and melt, and increasing global air temperatures will shift lake ice timing and diminish thickness [1,39,40]. While the magnitude of the trends in ice phenology and duration will vary based on the study, data used, and model type, the current most comprehensive collection of global lake ice records, extending from 1846 to 2019, indicates that warming air temperatures have resulted in delayed ice formation of 11.0 days and advanced breakup of 6.8 days [41]. Even while ice cover is diminishing in duration throughout the Northern Hemisphere, we found that warming lengthened the amount of time that ice cover is unsafe for use, even if it is present.

While ice cover is thickened by snow and lakes in regions with increased snow cover can show signs of increasing ice thickness [42], we found that air temperature governed the duration of the transition periods. When comparing CESM2-LE modeled results for air temperature, snow depth, and lake ice thickness, we found that thickness is overwhelmingly affected by air temperature rather than snow cover (S2 Fig). In areas throughout the Northern Hemisphere that are warming more, ice thickness decreased. The spatial patterns of unsafe transition period anomalies followed that behavior (Fig 3). Transition periods tended to increase most in areas where warming is expected, such as lower latitudes and coastal regions [1,16]. In fact, our derived transition period anomalies followed similar spatial patterns to both studies

that used CESM2-LE lake ice thickness data in other capacities [16,27] and those that used other methods to describe patterns in lake ice [1,43]. Areas that experience longer transition periods are in the minority; however, these regions can experiencing cooling temperatures [6].

Finally, changes in transition variability will impact the predictability of safe ice conditions. The increasing standard deviation in the transition period when temperatures increase and ice quality degrades indicates that lakes may have increasingly unpredictable safety conditions during ice cover periods as durations decrease and ice quality degrades [5,44]. Lake ice is typically safe in temperate Northern Hemisphere lakes in February when many people use lakes for recreation; however, accidents have occurred as lake ice becomes less predictable [5,15,40]. Lakes are projected to enter no-analogue scenarios as early as 2050 where average metrics such as lake temperature exceed current variability [27]. Regions in southern latitudes of North American, Europe and Asia have experienced intermittent ice cover between 2000–2009 and higher latitudes are expected to experience intermittent ice cover between 2090–2099 [16]. Additionally, lake ice duration has become more variable, especially when ice duration has significantly decreased [44].

## 4.2 Ice quality conditions predominantly dictate safe ice conditions

The transition period during which ice is present on a lake's surface, but unsafe for use, increases due to warming air temperatures; however, our findings show that changes to ice quality dramatically increase the length of the unsafe transition period when compared to an assumed black ice historical condition. Rising winter and spring air temperatures [39,41] increase proportions of white ice that compose the ice column [5], enhancing the variability of transition periods. For example, the average transition period of unsafe ice increases by 5.8 and 11 times longer because of ice quality degradation from 100% black ice to 50% white ice and 100% white ice, respectively, compared to 1.7 times longer for black ice conditions between 2°C and 4°C warming. Our results, then, suggest that ice quality is the primary concern for lake ice safety in the future. Warming air temperatures are the root cause of lake ice quality degradation [5]; however, the diminished density [10] and reduced flexural strength [9,10,45] that result from increasing proportions of white ice will lead to the most dangerous conditions when ice is present but unsafe, as noted in [5].

Increasingly degraded ice quality results in significantly longer mean transition period values and right-skewed transition period distributions, implying much longer maximum transition periods when ice quality degrades to greater proportions of white ice (Fig 4). We found that the distributions of transition periods across the Northern Hemisphere become increasingly skewed when accounting for the interdependence of warming and ice quality. While the Northern Hemisphere as a whole will tend toward longer average transition periods (see Table 1), lakes may have extremely long transition periods (i.e., longer than 1 month), shortening the viable ice cover period for activities more than the predicted ice duration loss of 38 ± 11 days [16], which used CESM2-LE ice thickness data. The additional loss of usable ice cover that the unsafe transition periods represent demonstrates that ice phenology metrics do not fully capture the overall loss of ice cover. While the presence of ice is important for ecosystems and biogeochemical processes [46–49], ice cover must be of sufficient thickness to be used for recreation and subsistence [15,21,50].

Climate warming is already enhancing variation in phenological events [51] and increasing the frequency of extreme events through significant shifts in mean ice timings [52], particularly in large lakes and warmer regions [3,52]. Lake ice cover tends to have increased variability in conjunction with climate oscillations, such as the North Atlantic Oscillation and the El Niño Southern Oscillation [44]. Therefore, longer transition periods may also respond to

shifting mean ice timings, lake morphology, local and regional climate, and climate oscillations. Furthermore, this increased variability leads to increasingly unpredictable ice safety as implied by the increasing standard deviations that occur with warming temperatures and degrading ice quality.

## 4.3 Consequences of increasingly unsafe ice

Lengthening transition periods of unsafe ice highlight the need to update and evolve safety protocols surrounding individual and community use of lake ice [50,53]. Ice conditions are shifting rapidly and ice loss has accelerated in recent decades [3]. These changes to ice conditions, including the phenology of lake ice, have contributed to dangerous conditions of ice during periods of human use [15]. The rapid decay of safe ice conditions suggests that the end of the ice season will offer the most dangerous conditions, which has been observed in an analysis of winter drowning data from the Northern Hemisphere [15]. Our results support this analysis, showing that while the ice formation and breakup periods occur later and earlier, respectively, the warming winter and spring seasons [54] prolong the amount of time between unsafe and safe ice thicknesses, despite the continued presence of ice.

Climate change not only shortens the ice season, it degrades the ice such that it is not as safe for human use. As an example, the Rideau Canal, which flows through Ottawa, Canada, is known to open annually for ice skating throughout the winter and is a site of historical and socio-ecological interest as the world's largest outdoor skating rink and a UNESCO World Heritage Site [55]. In the winter of 2023–2024, one of the warmest winters in recent history, the Canal opened for approximately one week before shutting down, owing to degraded and weak ice conditions, although the canal remained frozen. The weather did not reach the consecutive 10–14 days of -20 to -10°C temperatures, which are typical of January in Ottawa [56]. The weather tended to vacillate around 0°C alongside rain that made ice conditions unstable and unable to consistently reach the 30 cm required for safe operation, according to the National Capital Commission [56]. In the previous winter (2022–2023), the Rideau Canal did not open at all for the first time since the 1970–1971 winter. These adverse weather conditions have been increasing in frequency, which has shortened the duration of operation of the skateway by -5.2 ± 2.9 days between 1972 and 2013 [57], owing to warming weather and climate oscillations [58]. Though seasons can vary strongly (35–90 days), the period of skating days (seasonal length minus in-season closures) is projected to decline to 28±13.4 days by 2090 [57]. Other projections indicate shorter outdoor skating periods of 34% by 2090 in the Toronto area [59].

Beyond recreation, lake ice in Arctic regions is a necessity for mobility. Travel for Arctic communities involves navigating hundreds of kilometers of semi-permanent "trails" located on land, water bodies, lake ice, and sea ice [60]. Delayed ice formation or insufficient ice thickness or quality interrupts access to these trails. Trails are often used to access traditional foods, as hunting is seasonal, owing to migration, breeding, and feeding patterns [61]. For example, hunters may need to cross frozen lakes to access game; however, the timing of safe ice travel may conflict with seasonal hunting patterns. If alternate trails cannot be found (i.e., "trail switching") [60], communities may then have to cope with the shorter period or lack of hunting by increasing the ice fishing season, another activity that requires sufficient ice thickness and poses the risk of cold water submersion [62,63]. At the beginning of the season, the delay of sufficient ice thickness to use ice for fishing or hunting may require communities to increase fall fishing and hunting activities to cope with the inaccessibility of traditional winter hunting foods or rely on store-bought goods, though the window for those transportation periods is simultaneously shrinking [21,64]. While safe practices, such as Indigenous ice safety

knowledge and map-making (i.e., *Sikumik Qaujimajjuti*), can mitigate human risk on ice [65] and even mitigate impacts to trail access under climate change [60], changes to lake ice phenology and ice quality have strong impacts on the individuals and communities who rely on ice as a resource for transportation and food access.

The loss of safe lake ice conditions is a distinct reality for regions of the Northern Hemisphere. Our results indicate that lake ice will no longer be safe in more southern latitudes where lake ice may no longer grow thick enough for safe use in a warming world, especially when lake ice quality degrades and requires a thicker ice column to support human weight [5]. Lake ice is likely to become less consistent with warming air temperatures throughout the 21$^{st}$ Century, leading to intermittent ice cover in southern latitudes [66]. Moreover, as many as 5,700 lakes may also permanently lose ice cover with warming throughout the century [67]. Research using multiple model means, including the Community Land Model 4.5 (CLIM4.5), projects that lake ice thickness will decline by 3.3 cm per degree Celsius increase in air temperature [1]. An average loss of approximately 13.2 cm of ice thickness will make lakes at southern latitudes and coastal areas more likely to become unsafe, as their ice thickness is generally lower than northern latitudes [4]. These regions, therefore, should be particularly sensitive to the effects that changing ice thickness and quality have on human safety as the world warms.

## 5. Conclusion

The phenology of lake ice describes its first appearance and subsequent breakup; however, ice phenology does not adequately describe the use of lake ice. Where phenology in its broadest definition (i.e., ice on date and ice off date) is useful to describe the response of lakes to climate warming [40], the transition periods that we highlight in this study emphasize the role that timing has on the ecosystem services that communities derive from ice presence on lakes. The conjunction of warming and changing ice quality toward less stable white ice might lead to prolonged unsafe periods of 19.8±8.8 and 8.7±6.6 days for the ice formation and melt periods, respectively. Rapidly degrading ice conditions call for heightened vigilance during the ice breakup period [68]. A greater ratio of white ice earlier in the season will lead to unsafe ice conditions during periods when safe ice is anticipated by communities, given prior experience with safe ice timing [5]. This asynchronous relationship between actual ice safety and presumed ice safety from communities might lead to more frequent ice-related drownings [15]. Where ice presence is evident from a visual inspection, ice quality may require further scrutiny by drilling into the ice. Adaptation, then, is key in the face of rapidly shifting ice conditions [60], and improved safety protocols must be included for winter ice use during recreation and ice use broadly [65] to prevent winter drownings.

## Supporting information

**S1 Fig. Observed and modeled maximum ice thickness.** A comparison of modeled and observed data for the maximum ice thickness for each of the 71 validation lakes. Each point represents an individual lake. The line is the regression line with the R2 and line equation in the upper left of the plot.
(TIF)

**S2 Fig. Lake ice variability from climate forces.** The anomaly of ice thickness comapred to a historical condition (1851–1880) (colored points) when considering the forcing of air temperature anomalies (y-axis) and snow depth anomalies (x-axis). Each point represents the mean of an individual grid cell at 4°C warming.
(TIF)

**S3 Fig. Unsafe transition period empirical cumulative distribution functions.** The empirical cumulative distribution functions (ECDF) of each combination of warming scenarios and ice quality scenarios. Represented are (a) the ECDFs of each warming scenario (colored lines for 1°C, 2°C, and 4°C) with ice quality held constant. The left columns of both (a) and (b) show the ice formation transition period (i.e., "on") and the right columns show the ice melt transition period (i.e., "off").
(TIF)

**S4 Fig. Unsafe transition period empirical cumulative distribution functions.** The same as in S3 Fig but represented are the ECDF of each ice quality scenario (100% black ice, 50% black ice– 50% white ice, and 100% white ice) with warming held constant.
(TIF)

**S1 Table. Shapiro-Wilks result table.** The results of the Shapiro-Wilks test for each ice quality and warming scenario.
(PDF)

**S2 Table. Kruskal-Wallis results table comparing quality scenarios.** The results of the Kruskal-Wallis rank sum test, which compared the ice formation and melt transition period anomalies by the lake ice quality scenario (i.e., 100% black ice, 50%black/white ice, 100% white ice).
(PDF)

**S3 Table. Kruskal-Wallis results table comparing warming scenarios.** The results of the Kruskal-Wallis rank sum test, which compared the ice formation and melt transition period anomalies by the warming scenario (i.e., 1°C, 2°C, 4°C).
(PDF)

**S4 Table. Comparing across ice quality and warming scenarios with Dunn's test.** The results from the Dunn's test of multiple comparison using rank sums within each ice group (black, white, etc.) across warming scenarios (i.e., 1°C, 2°C, 4°C). The adjusted p value is provided, using the Holm method for detecting the family-wise error rate.
(PDF)

**S5 Table. Comparing across warming scenarios and ice quality with Dunn's test.** The results from the Dunn's test of multiple comparison using rank sums within each warming scenarios (i.e., 1°C, 2°C, 4°C) across ice group (black, white, etc.). The adjusted p value is provided, using the Holm method for detecting the family-wise error rate.
(PDF)

**S6 Table. Comparing warming scenarios across ice quality categories.** The results from the asymptotic two-sample Kolmogorov-Smirnov test. The compared samples are within the same ice quality scenarios (i.e., 100% black ice, 100% white ice, and 50% white ice) and across warming scenarios (i.e., 1°C, 2°C, 4°C).
(PDF)

**S7 Table. Comparing ice quality across warming categories.** The results from the asymptotic two-sample Kolmogorov-Smirnov test. The compared samples are within the same warming scenario (i.e., 1°C, 2°C, 4°C) and across ice quality scenarios (i.e., 100% black ice, 100% white ice, and 50% white ice).
(PDF)

## Acknowledgments

We would like to thank the Arctic Mobilities team, including Alison Cook, Jackie Dawson, Chris Derksen, Stephen Howell, and Lawrence Mudryk, for extensive discussions and critical feedback on this research. We would also like to thank Aman Basu for assistance with data processing. Finally, we thank the Academic Editor and two anonymous reviewers for their constructive comments and helpful feedback. The CESM2-LE simulations presented here have been conducted through a partnership between the Institute for Basic Sciences (IBS) Center for Climate Physics (ICCP) in South Korea and the Community Earth System Model (CESM) group at the National Center for Atmospheric Research (NCAR) in the United States, representing a broad collaborative effort between scientists from both centers. The simulations were conducted on the IBS/ICCP supercomputer "Aleph".

## Author Contributions

**Conceptualization:** Joshua Culpepper, R. Iestyn Woolway, Sapna Sharma.

**Data curation:** Joshua Culpepper, Lei Huang, R. Iestyn Woolway.

**Formal analysis:** Joshua Culpepper, Lei Huang, R. Iestyn Woolway, Sapna Sharma.

**Funding acquisition:** R. Iestyn Woolway, Sapna Sharma.

**Investigation:** Joshua Culpepper, R. Iestyn Woolway, Sapna Sharma.

**Methodology:** Joshua Culpepper, Lei Huang, R. Iestyn Woolway.

**Supervision:** R. Iestyn Woolway, Sapna Sharma.

**Validation:** Lei Huang.

**Visualization:** Joshua Culpepper, R. Iestyn Woolway, Sapna Sharma.

**Writing – original draft:** Joshua Culpepper, Lei Huang, R. Iestyn Woolway, Sapna Sharma.

**Writing – review & editing:** Joshua Culpepper, Lei Huang, R. Iestyn Woolway, Sapna Sharma.

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
