## [Decision Letter · Decision Letter 0]

1 Jul 2024

PONE-D-24-18915Widespread loss of safe lake ice access in response to a warming climatePLOS ONE

Dear Dr. Culpepper,

Thank you for submitting your manuscript to PLOS ONE. After careful consideration, we feel that it has merit but does not fully meet PLOS ONE’s publication criteria as it currently stands. Therefore, we invite you to submit a revised version of the manuscript that addresses the points raised during the review process.

We look forward to receiving your revised manuscript.

Kind regards,

Sher Muhammad, PhD

Academic Editor

PLOS ONE

“The Natural Sciences and Engineering Research Council Discovery Grant, the York University Research Chair programme, and ArcticNet provided funding to Sapna Sharma to support this research.

R. Iestyn Woolway was supported by a UKRI Natural Environment Research Council (NERC) Independent Research Fellowship [grant number NE/T011246/1].

Lei Huang was supported by the National Natural Science Foundation of China (No. 42201049) and the Second Tibetan Plateau Scientific Expedition and Research (STEP) (grant number 2019QZKK0202).”

“We would like to thank the Arctic Mobilities team for extensive discussions over the past year, including Alison Cook, Jackie Dawson, Chris Derksen, Stephen Howell, and Lawrence Mudryk. We would also like to thank the Natural Sciences and Engineering Research Council Discovery Grant, the York University Research Chair programme, and ArcticNet to SS for providing funding to support this research. RIW was supported by a UKRI Natural Environment Research Council (NERC) Independent Research Fellowship [grant number NE/T011246/1]. Lei Huang was supported by the National Natural Science Foundation of China (No. 42201049) and the Second Tibetan Plateau Scientific Expedition and Research (STEP) (grant number 2019QZKK0202). The CESM2-LE simulations presented here have been conducted through a partnership between the Institute for Basic Sciences (IBS) Center for Climate Physics (ICCP) in South Korea and the Community Earth System Model (CESM) group at the National Center for Atmospheric Research (NCAR) in the United States, representing a broad collaborative effort between scientists from both centers. The simulations were conducted on the IBS/ICCP supercomputer “Aleph.”

“The Natural Sciences and Engineering Research Council Discovery Grant, the York University Research Chair programme, and ArcticNet provided funding to Sapna Sharma to support this research.

R. Iestyn Woolway was supported by a UKRI Natural Environment Research Council (NERC) Independent Research Fellowship [grant number NE/T011246/1].

Lei Huang was supported by the National Natural Science Foundation of China (No. 42201049) and the Second Tibetan Plateau Scientific Expedition and Research (STEP) (grant number 2019QZKK0202).”

Additional Editor Comments:

The manuscript has been reviewed by two anonymous reviewers. The study carry out a significant step forward in predicting lake ice safety in a warming climate, with important findings on how ice quality impacts safety and how warming affects the duration of safe and unsafe ice periods. However, some improvements are required such as terminology needs to be more precise, with clearer definitions for key terms. The abstract and introduction should provide more context and move detailed climatic information from the discussion to the introduction. Model validation should be expanded with more detailed statistical analysis and inclusion of validation figures in the main text. The results and discussion sections need deeper exploration of model predictions and drivers of variability. I hope the revision =will strengthen the quality, readability, precision, and validation.

Reviewers' comments:

Reviewer's Responses to Questions

**Comments to the Author**

1. Is the manuscript technically sound, and do the data support the conclusions?

Reviewer #1: Partly

Reviewer #2: Yes

2. Has the statistical analysis been performed appropriately and rigorously? 

Reviewer #1: No

Reviewer #2: No

3. Have the authors made all data underlying the findings in their manuscript fully available?

Reviewer #1: No

Reviewer #2: No

4. Is the manuscript presented in an intelligible fashion and written in standard English?

Reviewer #1: Yes

Reviewer #2: Yes

5. Review Comments to the Author

Reviewer #1: This paper addresses an important and interesting problem: lake ice phenology and quality to better assess safe lake ice use during the formation and melt periods.

1) This paper described that “Lake ice on timing was validated against 328 lakes from the Global Lake and River Ice Phenology database with >20 years of data (20). The correlation coefficient r was 0.78 with a mean absolute error (MAE) of 11.7 days. Ice-off timing had an r = 0.94 with an MAE of 15.6 days from a sample of 322 lakes. Ice thickness was validated with records from 15 Canadian lakes with an r = 98 0.93”. (1) it needs to provide detailed information about spatio-temporal distribution of lake ice-on and ice-off timing and lake ice thickness. (2) The correlation coefficient r was 0.78 with a mean absolute error (MAE) of 11.7 days. Ice-off timing had an r = 0.94 with an MAE of 15.6 days from a sample of 322 lakes. There are a large uncertainties of research results like “… The transition period of unsafe ice increases by 4.97 ± 3.67 days in a 4 °C warmer world…”,” …The unsafe transition 33 period increases by an average of 19.8 ± 8.84 days and 8.75 ± 6.63 days…” according to MAE (Mean Absolute Error). How to proof the accuracy of results?

2) The introduction could be enhanced by adding relevant studies on the state of lake ice cover in the Northern Hemisphere, and even globally, under the background of climate warming. This would provide some research background support for this study.

3) It is recommended to add a discussion section about the prospects of the paper, discussing whether future research could propose opinions on regional lake ice safety based on ice thickness. For example, which areas have feasible ice safety and which do not.

4) It is suggested to divide the methods and results sections into paragraphs, such as making the methods the second section (2. Methods, 2.1, 2.2, 2.3, etc.), to make the research process clearer and easier for readers to understand. Adding a graphical abstract would be even better.

Reviewer #2: This manuscript presents a model study of lake ice phenology and quality changes associated with warmer air temperatures. It represents a critical step forward toward predicting the safety of lake ice in a warmer world. The model predictions indicate that in addition to shorter ice cover periods, the period from unsafe ice to safe ice is lengthening. An important result is that ice quality can be the dominant determinant of ice safety. However, I find that some of the terms could be defined more precisely, and model validation could be improved.

General comments:

Abstract:

- Please include a brief explanation of how ice quality impacts ice safety.

- Please describe briefly the analysis to contextualize the changes in the transition period. For example, the scope of the analysis is unclear; how many lakes (or grid cells with lakes) were modelled?

- The language is vague in places: ‘up to 35 days longer in a warming world’. What does ‘a warming world’ mean quantitatively?

Introduction:

- I miss a paragraph explaining the formation and properties of black versus white ice. Why is white ice considered less safe?

- I also miss a paragraph briefly describing the anticipated or ongoing climatic changes that affect lake ice formation, such as rising winter air temperatures, changing wind speed and a shift in the type and the amount of precipitation. OK, I see now that some of this information is included in the Discussion (L276-289). I would recommend moving this paragraph to the introduction.

Methods:

- If I understand correctly, the authors used the CESM2-LE + LISSS model to obtain predictions of ice thickness at various climate warming scenarios, and then computed the length of the ‘unsafe’ and ‘safe’ ice periods based on ice thickness recommendations from the MDNR and the assumption that the ice is either 100% black ice, 50% black ice and 50% white ice, or 100% white ice.

- Outcomes of statistical tests should include the analysis and the sample size (e.g. L96, r = 0.92).

- Why is the Pearson correlation coefficient (r) used here and not the coefficient of determination (r2), which is usually used to determine model fit?

- The model validation could be expanded upon. For example, you can have a high coefficient of determination, but a significant deviation from the 1:1 line. I suggest that figures showing the model validation be included in the main text. I would also suggest that the authors look for a larger and more representative lake ice thickness dataset to validate the model.

Results

- A key part of the results is a comparison between modelled transition times from ice formation to ‘safe’ ice for both black ice and white ice. How realistic is the immediate formation of white ice after ice-on, given what we know about the formation process of white ice? For example, does the model predict more freeze-thaw cycles at the start of the season? If it is not a realistic scenario, is it useful?

- The regional distribution of transition times is not mentioned in the results, but it appears that some regions are transitioning to longer ‘safe’ ice seasons, correct?

Discussion

- The discussion seems to be a combination of introduction and results. I’d like to see a deeper exploration as to why the model predicts the shifts in the transition times (Fig. 2) and what variable(s) determine(s) their global distribution (Fig. 3). Some of this information should be in the model results. For example, are the number of freeze-thaw cycles increasing? Is the decline in the ice growth rate due to the direct influence of warmer air temperatures or also indirect effects, such as increasing precipitation as snow (insulation) or rain (melt)?

- Figure 2 is not discussed in much detail. What are the drivers of the spatial variability in transition period trends?

Specific comments:

L29: doesn’t the safe ice thickness depend on the ratio of black and white ice?

L31-33. ‘4.97 ± 3.67 days’ given the confidence interval (standard deviation?) it may be more appropriate to use one decimal here.

L36. ‘increase the risk of drowning’. Is this substantiated?

L37-38: ‘when black ice transitions to white ice conditions’. Doesn’t this usually happen at the end of winter?

L50: only North America?

L57: what is meant by ‘stable’?

L62. Here or earlier, please provide a working definition of ‘safe ice’ (e.g. safe for one person to walk on, 5 cm for black ice and 10 cm for white ice) – with a reference (e.g. ref. 7 or Gold 1971).

L97-98: why were only 15 lakes used to validate lake ice thickness?

L102: What does ‘weaker’ mean here? Barrette (ref. 21) estimated the flexural strength of ice, which is proportional, but not identical to load-bearing strength. However, their results roughly correspond to those of Gold 1971, who found that the load-bearing capacity of white ice was about 50% that of black ice based on documented ice failure under vehicular loads.

L108: please add that the ice quality gradients are based on assumptions/scenarios of 100% black ice, 50% black ice and 50% white ice, and 100% white ice.

L110: why is 2 cm used as the threshold for ice formation, and not >0 cm?

L112: What load (e.g. a person, or a car) do these safety recommendations apply to?

L117: Why these temperatures? Do they correspond to specific climate warming scenarios?

L119: were some grid cells excluded from the analysis? These could be grid cells without lakes or without freezing days. If so, please describe the selection procedure.

L121-122: Please provide a (supplementary) table showing the outcomes of all your statistical tests, including the test statistic and the sample size.

L121-122: ‘Shaprio-Wilks’ -> ‘Shapiro-Wilks’

L121-122: why is the Shapiro-Wilks test done when the other statistical tests are non-parametric (i.e. do not require the assumption of a particular distribution)?

L124: ‘Kurskal-Wallis’ -> ‘Kruskal-Wallis’

L124: conventionally (and arbitrarily), p < 0.05 is used. Why this threshold?

L126: ‘Kalmogorov-Smirnov’ -> ‘Kolmogorov-Smirnov (KS)’

L128: is this a two-sample comparison or a multiple comparison? If the latter, it is necessary to adjust p for the false discovery rate using the Benjamini-Hochberg or equivalent procedure.

L162-163: please define the uncertainty estimates (SD, 95% confidence interval) and the sample size (n = …).

L165-166: suggests changing ‘(≥10 cm, ≥15 cm, and ≥20 cm)’ to ‘(≥10 cm, ≥15 cm, and ≥20 cm, depending on assumed ice quality)’.

L179: Table 1, please include the outcome of the statistical test indicating whether or not each mean is significantly different from 0 at a specified confidence level (e.g. p = 0.05), or show the p-values.

L193: mention that you are talking about 100% black ice here.

L219: Figure 3 caption. when mentioning p-values it is customary to also name the statistical test and the sample size.

L233-244: this reads as part of the introduction. No results are discussed here as far as I can tell.

L232: ‘forecast’ is a term not usually applied to short-term rather than long-term model predictions. ‘Project’ or ‘Predict’ may be more appropriate here.

L245: suggest changing ‘forecasted’ to ‘projected’.

L246: did you mean ‘the primary mechanism through which we understand changes in ice phenology and duration…’?

L248: ‘some of the most recent long-term trends’: what is meant here? Trends since the year 2000?

L252: please provide a bit more context for these previous studies. ‘ice thickness declines’; ice thickness varies over the season, so is this the mean ice thickness? A 1C rise in air temperature; is this the annual mean air temperature? And did these previous studies cover the same lakes/area as this study?

L252: days C−1 -> days °C−1

L253: ‘projections forecast’: please see my previous comment on the use of ‘forecast’.

L257-258: which scenarios are compared here? (e.g. historical formation of black ice vs a future formation of white ice at 4°C).

L260: 9 days earlier. Do you mean ‘9 days before ice-off’ compared to ‘4 days before ice-off historically’? If so, again, which scenario is this prediction associated with? Also, this is a result which should be in the ‘Results’ section.

L276-289: this paragraph should be in the introduction. No results are discussed here.

L290: please provide references for the statements ‘rising winter and spring air temperatures’ and ‘increasing proportions of white ice’.

L299: this was already stated on L297.

L304-305: is the 29 days an average for all the lakes at 4 °C? If so, is this the same modelling approach with the same lakes (i.e. are the results comparable)?

L310: can the cumulative distributions be shown in a (supplementary) figure?

L317: ‘increasingly unpredictable ice safety’. This is an important result. Could this be quantified? E.g. compare the standard deviation of the transition period length (not the anomaly) between historic and warming scenarios and see if it increases.

L346: this paragraph is a combination of methods and results. Please restructure.

L392: ‘which some individuals may fail to take into account’. This is unscientific speculation, please remove.

6. PLOS authors have the option to publish the peer review history of their article (what does this mean?). If published, this will include your full peer review and any attached files.

Reviewer #1: **Yes: **Yusufujiang Rusuli

Reviewer #2: No

---

## [Author Response · Author response to Decision Letter 0]

3 Sep 2024

All of the responses to reviewers #1 and #2 are noted within the marked copy of the manuscript as well as the "Response to Reviewers" Word Document, which has been uploaded as a document to this submission. Below I will copy the text from that document.

Before I paste that text, I would like to thank both reviewers for their thorough and useful comments. Those comments have generated a much improved manuscript. 

Editor Comments: 

Comment: The manuscript has been reviewed by two anonymous reviewers. The study carry out a significant step forward in predicting lake ice safety in a warming climate, with important findings on how ice quality impacts safety and how warming affects the duration of safe and unsafe ice periods. 

Response: Thank you for considering our manuscript publication and the helpful revision summary. Please, see the individual comments and marked document for all of the responses and changes to the initial manuscript submission. 

Comment: However, some improvements are required such as terminology needs to be more precise, with clearer definitions for key terms. 

Response: Using the reviewers’ suggestions, we have added important context to the introduction and we have ensured that key terms are clearly defined throughout the manuscript. 

Comment: The abstract and introduction should provide more context and move detailed climatic information from the discussion to the introduction. 

Response: We have worked to add the needed context for the abstract and introduction. The specific changes can be seen in the responses to the reviewers. 

Comment: Model validation should be expanded with more detailed statistical analysis and inclusion of validation figures in the main text. 

Response: We agree and have expanded the validation to now include 71 ice thickness records between 1980-present. We found that the RMSE is 18 cm and the R2 = 0.74, where the modeled data tend to underestimate the observed peak ice thickness, but the ice growth is predicted within one standard deviation of the model mean. We added a figure comparing both the modeled and observed lake ice thickness for all lakes in the dataset alongside a time series that shows the variation between the modeled and observed data for one of the studied lakes. For more details and added text, please, see the responses to reviewers 1 and 2. 

Comment: The results and discussion sections need deeper exploration of model predictions and drivers of variability. I hope the revision will strengthen the quality, readability, precision, and validation. 

Response: We have worked to revise these sections with the feedback from reviewers 1 and 2. We offer additional information within the results section on spatial variability and its drivers as well as more discussion to synthesize these results in section 4.1. 

Reviewer #1: 

Comment: This paper addresses an important and interesting problem: lake ice phenology and quality to better assess safe lake ice use during the formation and melt periods. 

Response: Thank you for noting the importance of this manuscript dealing with what we see as a critical component of lake ice and cryosphere studies. 

Comment: 1) This paper described that “Lake ice on timing was validated against 328 lakes from the Global Lake and River Ice Phenology database with >20 years of data (20). The correlation coefficient r was 0.78 with a mean absolute error (MAE) of 11.7 days. Ice-off timing had an r = 0.94 with an MAE of 15.6 days from a sample of 322 lakes. Ice thickness was validated with records from 15 Canadian lakes with an r = 98 0.93”. 

(1) it needs to provide detailed information about spatio-temporal distribution of lake ice-on and ice-off timing and lake ice thickness. 

Response: We have indeed additional context to the introduction regarding lake ice timing and thickness. We have also added a section to the results that describes the spatio-temporal distribution of transition periods. We added the following to better describe these aspects of the results. You can also view our response to reviewer 2’s inquiry regarding the spatial variability of grids. 

“Transition periods show distinct regional variability throughout the Northern Hemisphere but overwhelmingly shift toward longer transition periods. Unsafe transition periods on average increased (Table 1; Figure 3) and transition periods tended to increase the most along coastal regions in North America as well as southern latitudes in North America, Europe, and Asia. For example, transition periods exceeding 10 days composed approximately 30% of grids (n = 661) for white ice conditions at 4 °C. On the other hand, grids shifting to longer transition periods consisted of less than 1% of grids (n = 14). More grids shift toward longer transition periods at 1 C under black ice conditions (18%, n = 496). These grids are primarily located in northern latitudes of North America, Europe, and Siberia as well as some grids along the western coast of Greenland.” 

(2) The correlation coefficient r was 0.78 with a mean absolute error (MAE) of 11.7 days. Ice-off timing had an r = 0.94 with an MAE of 15.6 days from a sample of 322 lakes. There are a large uncertainties of research results like “… The transition period of unsafe ice increases by 4.97 ± 3.67 days in a 4 °C warmer world…”,” …The unsafe transition 33 period increases by an average of 19.8 ± 8.84 days and 8.75 ± 6.63 days…” according to MAE (Mean Absolute Error). How to proof the accuracy of results? 

Response: We have significantly revised the methods section to more accurately tailor to the research that we conducted. Initially, we highlighted the validation work conducted in Huang et al., 2022 who conducted analyses regarding lake ice thickness. We have revised the methods to focus on lake ice thickness results by increasing the validation lakes from 15 to 71. We included the following to the methods section to fully describe the validation process: 

“The spatial extent and resolution of the study include the Northern Hemisphere on a 0.9°-by-1.25° gridded scale. Within each grid, the LISSS coupled to the CESM2-LE uses a sample lake to model ice thickness, where the sample lake has a depth and surface area of the mean of individual lakes within the grid, taken from the Global Lake and Wetland Database (22) and global gridded depth data at a 1 km resolution (23). CESM2-LE offers data for lake ice-on and ice-off timing, using the daily lake ice thickness from the one-dimensional lake ice model. The LISSS model has shown accurate representation of ice thickness and other variables (e.g., water temperature and surface fluxes) in previous studies (16,20,24–27). Ice thickness was validated with records from 15 Canadian lakes with an R2 = 0.86. While the correlation coefficient was high, the modeled data deviated from the 1:1 line, such that the model tended to underpredict ice thickness (slope = 2.1) (16). This bias toward thinner ice cover values is likely due to the warm bias of the simulated land surface air temperature in the fully coupled model, which was warmer than the ERA5 reanalysis data between the study years 1981-2020 and 30°N-70°N latitude by 1.5 °C (16). Additional validation information can be found in (16) and the associated supplementary information. 

Owing to the small sample size in (16), we sought to further validate the ice thickness component of LISSS. We aggregated 71 lakes from datasets from Finland (28), North America (29,30), and Russia (31). Using 71 ice thickness records with data between 1980-present, ice thickness has a root mean square error (RMSE) of 18.5 cm and linear regression shows an R2 = 0.74 (n = 71) (Figure 1). As with the validation procedure in (16), the modeled data tended to underpredict ice thickness, which is evident from the slope of 0.49 (Figure 1a). When looking directly at lake ice thickness time series, the mean ice thickness of the ensemble members tends to underestimate the peak ice thickness (Figure 1b). The mean and standard deviation may also underpredict the peak ice thickness; however, they tend to capture the ice growth period and ice loss period, which are most important for this study. The model also tends to predict lake ice every year (1980-2024); however, some lakes did not freeze in years during data collection. It should be noted that the lake ice thickness model (LISSS) uses a mean lake area and depth to model ice thickness (16,20). Therefore, peak lake ice thickness is likely to vary between lakes that do not approximate that mean area and depth. ” 

Comment: 2) The introduction could be enhanced by adding relevant studies on the state of lake ice cover in the Northern Hemisphere, and even globally, under the background of climate warming. This would provide some research background support for this study. 

Response: We revised the first paragraph of the introduction to add context regarding the state of lake ice cover in the Northern Hemisphere. The revised paragraph now reads: 

“Anthropogenic climate change is causing rapid loss of lake ice in the Northern Hemisphere, driven largely by rising air temperatures (1). Studies have illustrated that the timing of ice-on and ice-off (ice phenology) is changing around the Northern Hemisphere. Lake ice duration has been a central concern, where long-term records have shown an overall increase in open-water days of 0.62 days per decade between 1931-2005, with a nonlinear increase in the rate of open-water days in the last 30 years of the time series (2). Lake ice phenology through long-term records extending over a century (1846-2019) show ice formation to be 11 days later and ice breakup to be 6.8 days earlier, which was a near doubling of ice loss compared to the same lake ice records between 1846-1995 (3). Therefore, long-term lake ice records show an overall loss of ice cover duration, timing, and thickness directly related to warming air temperatures (1,4).” 

Comment: 3) It is recommended to add a discussion section about the prospects of the paper, discussing whether future research could propose opinions on regional lake ice safety based on ice thickness. For example, which areas have feasible ice safety and which do not. 

Response: We have added a discussion section to address the prospects of the paper. We focus on areas that are of highest concern for lake ice safety as well as why certain regions may no longer have safe ice in the future. We have added the following to discussion section 4.3. 

“The loss of safe lake ice conditions is a distinct reality for regions of the Northern Hemisphere. Our results indicate that lake ice will no longer be safe in more southern latitudes where lake ice may no longer grow thick enough for safe use in a warming world, especially when lake ice quality degrades and requires a thicker ice column to support human weight (5). Lake ice is likely to become less consistent with warming air temperatures throughout the 21st Century, leading to intermittent ice cover in southern latitudes (67). Moreover, as many as 5,700 lakes may also permanently lose ice cover with warming throughout the century (68). Research using multiple model means, including the Community Land Model 4.5 (CLIM4.5), projects that lake ice thickness will decline by 3.3 cm per degree Celsius increase in air temperature (1). An average loss of approximately 13.2 cm of ice thickness will make lakes at southern latitudes and coastal areas more likely to become unsafe, as their ice thickness is generally lower than northern latitudes (4). These regions, therefore, should be particularly sensitive to the effects that changing ice thickness and quality have on human safety as the world warms.” 

Comment: 4) It is suggested to divide the methods and results sections into paragraphs, such as making the methods the second section (2. Methods, 2.1, 2.2, 2.3, etc.), to make the research process clearer and easier for readers to understand. Adding a graphical abstract would be even better. 

Response: We have included the section titles as recommended throughout the document for each section. 

Reviewer #2: 

Comment: This manuscript presents a model study of lake ice phenology and quality changes associated with warmer air temperatures. It represents a critical step forward toward predicting the safety of lake ice in a warmer world. The model predictions indicate that in addition to shorter ice cover periods, the period from unsafe ice to safe ice is lengthening. An important result is that ice quality can be the dominant determinant of ice safety. However, I find that some of the terms could be defined more precisely, and model validation could be improved. 

Response: Thank you for your brief summation of the paper, which helps confirm that the key points were accessible. We also thank you for the thorough review, including the more precise defining of terms and enhancing the model validation. 

General comments: 

Abstract: 

Comment: Please include a brief explanation of how ice quality impacts ice safety. 

Response: We appreciate that a brief explanation of ice quality and safety can help the impact of the abstract. We have added the sentence, “Later formation and earlier breakup of lake ice lead to overall short periods of use, but greater proportions of white ice, which may inhibit safe ice use due to its lower weight-bearing capacity.” 

Comment: Please describe briefly the analysis to contextualize the changes in the transition period. For example, the scope of the analysis is unclear; how many lakes (or grid cells with lakes) were modelled? 

Response: We used the Community Earth System Model version 2 Large Ensemble (CESM2-LE) to calculate the period when ice first appears until it is of sufficient thickness for safe use, depending on the ratio of black to white ice. We conducted this analysis for between 2,379-2,829 ~1°x1° grids throughout the Northern Hemisphere. We added the following to the abstract: “We used the Community Earth System Model Version 2 Large Ensemble (CESM2-LE) to calculate the period when ice first appears until it is of sufficient thickness for safe use, which depends on the ratio of black to white ice. We conducted this analysis ranging from 2,379 to 4,829 ~1° by 1° grid cells throughout the Northern Hemisphere. We focus on the period between ice formation (≥ 2 cm) to a safe thickness for general human use (i.e., ≥10, ≥15, or ≥20 cm, depending on the ratio of black to white ice).” 

Comment: The language is vague in places: ‘up to 35 days longer in a warming world’. What does ‘a warming world’ mean quantitatively? 

Response: We chose to include both the low (black ice) and high (white ice) transition period boundaries for the average increases in the transition periods rather than just the white ice conditions. We also added the specific temperature increase that led to those values. 

We have amended the sentence to read, “We show that although many lakes are forecasted to freeze, they will be unsafe to use for between 5 and 29 fewer days in a 4 °C warmer world for black and white ice conditions, respectively.” 

Introduction: 

Comment: I miss a paragraph explaining the formation and properties of black versus white ice. Why is white ice considered less safe? 

Response: Please see the next comment for the full revision of the introductory paragraphs. 

Comment: I also miss a paragraph briefly describing the anticipated or ongoing climatic changes that affect lake ice formation, such as rising winter air temperatures, changing wind speed and a shift in the type and the amount of precipitation. OK, I see now that some of this information is included in the Discussion (L276-289). I would recommend moving this paragraph to the introduction. 

Response: We agree that paragraphs explaining the formation and properties of black versus white ice (comment above) and how climatic changes will help better contextualize the research as a whole. We have revised the introduction to include a discussion of the current condition of lake ice broadly (based on 

---

## [Decision Letter · Decision Letter 1]

5 Nov 2024

Widespread loss of safe lake ice access in response to a warming climate

PONE-D-24-18915R1

Dear Dr. Culpepper,

We’re pleased to inform you that your manuscript has been judged scientifically suitable for publication and will be formally accepted for publication once it meets all outstanding technical requirements.

Kind regards,

Sher Muhammad, PhD

Academic Editor

PLOS ONE

Additional Editor Comments (optional):

The revision has improved the quality of the manuscript, I recommend it for publication. The authors are also suggested to made the data available including the validation data.

Reviewers' comments:

Reviewer's Responses to Questions

**Comments to the Author**

1. If the authors have adequately addressed your comments raised in a previous round of review and you feel that this manuscript is now acceptable for publication, you may indicate that here to bypass the “Comments to the Author” section, enter your conflict of interest statement in the “Confidential to Editor” section, and submit your "Accept" recommendation.

Reviewer #2: All comments have been addressed

2. Is the manuscript technically sound, and do the data support the conclusions?

Reviewer #2: Yes

3. Has the statistical analysis been performed appropriately and rigorously? 

Reviewer #2: Yes

4. Have the authors made all data underlying the findings in their manuscript fully available?

Reviewer #2: No

5. Is the manuscript presented in an intelligible fashion and written in standard English?

Reviewer #2: Yes

6. Review Comments to the Author

Reviewer #2: I appreciate the effort by the authors to address all comments. The manuscript reads much clearer now.

Please ensure that the model calibration data (ice thickness measurements from 71 lakes) are also available; I did not see this in the data availability statement.

7. PLOS authors have the option to publish the peer review history of their article (what does this mean?). If published, this will include your full peer review and any attached files.

Reviewer #2: No

---

## [Editor Report · Acceptance letter]

8 Nov 2024

PONE-D-24-18915R1 

PLOS ONE

Dear Dr. Culpepper, 

I'm pleased to inform you that your manuscript has been deemed suitable for publication in PLOS ONE. Congratulations! Your manuscript is now being handed over to our production team.

Kind regards, 

on behalf of

Dr. Sher Muhammad 

Academic Editor

PLOS ONE